# Measuring Progress in Deep Reinforcement Learning Sample Efficiency

## Abstract

Sampled environment transitions are a critical input to deep reinforcement learning (DRL) algorithms. Current DRL benchmarks often allow for the cheap and easy generation of large amounts of samples such that perceived progress in DRL does not necessarily correspond to improved sample efficiency. As simulating real world processes is often prohibitively hard and collecting real world experience is costly, sample efficiency is an important indicator for economically relevant applications of DRL. We investigate progress in sample efficiency on Atari games and continuous control tasks by comparing the number of samples that a variety of algorithms need to reach a given performance level according to training curves in the corresponding publications. We find exponential progress in sample efficiency with estimated doubling times of around 10 to 18 months on Atari, 5 to 24 months on state-based continuous control and of around 4 to 9 months on pixel-based continuous control depending on the specific task and performance level.

## 1 Introduction

Recent successes of deep reinforcement learning (DRL) in Go (Silver et al., 2016; 2017; 2018) and complex real-time strategy games (Berner et al., 2019; Vinyals et al., 2019) indicate the vast potential for automating complex economically relevant tasks like assembling goods in non-standardized environments or physically assisting humans using DRL. However, sample efficiency is likely to remain an important bottleneck for the economic feasibility of real world applications of DRL. This is because samples of state transitions in the environment caused by an agent's action are essential for training DRL agents but automated systems interacting with the real world are often fragile, slow or costly, which makes DRL training in the real world expensive both in terms of money and time (Dulac-Arnold et al., 2019). As most widely used benchmarks in DRL still consist of computer games and simulations where samples can be obtained risk-free, fast and cheaply[1], progress in DRL does not necessarily translate to future real world applications unless it corresponds to progress in sample efficiency or simulation accuracy and transfer learning. This makes information about progress in sample efficiency an important input to researchers studying topics like the future of employment (Frey & Osborne, 2017), the potential for malicious uses of DRL (Brundage et al., 2018) and other potentially transformative impacts of AI systems in the future (Gruetzemacher & Whittlestone, 2019).

## 2 Background and related work

### 2.1 Sample efficiency

While research on progress in the field of AI used to have a strong focus on benchmark performance metrics in AI (Eckersley & Nasser, 2017), there have been calls to pay more attention to the importance of other metrics: Martinez-Plumed et al. (2018) enumerate previously neglected dimensions of AI progress and explore the relationship between computing resources and final performance in RL. The wider implications of sample efficiency, more specifically, are explored by Tucker et al. (2020). While their scope is broader than DRL, most of their points do apply to reinforcement learning as a

---

[1]Espeholt et al. (2019) reach up to 2.4 M FPS and 1 B frames per 25$ on DM Lab (Beattie et al., 2016).

special case of AI technology. For example, as the authors point out, robotics is an especially important case of a data-limited domain, in which new applications are likely to be unlocked for many actors with improved sample efficiency. While some benchmark papers make explicit quantitative comparisons of sample efficiency (or "data efficiency") to isolated previous methods (Hessel et al., 2018; Hafner et al., 2019a; Kaiser et al., 2019; Srinivas et al., 2020), we are not aware of any more systematic investigation of sample efficiency in DRL.

## 2.2 COMPUTING POWER AND SCALING LAWS

While DRL and other applications of deep learning have produced many impressive results, this has gone hand in hand with an ever-increasing amount of computing power (compute) being devoted to model training: Compute usage for large AI projects has been in line with Moore's law prior to 2012 (Sastry et al., 2019), but saw a doubling time of 3.4-months in the 2012 to 2018 period, which is seven times faster than Moore's law (Amodei & Hernandez, 2018). Compute usage would not increase if there was nothing to gain: For example, the performance of Transformer-based language models (Vaswani et al., 2017) follows a power law in the number of model parameters when differently sized models are trained on the same dataset indicating that in the current regime, Transformer language models become more sample efficient with more parameters. Data is still important, as there is another power law relating dataset size for models with the same amount of parameters to performance (Kaplan et al., 2020). Things likely look similar for the ImageNet benchmark (Deng et al., 2009) where progress has long been accompanied by larger models using a fixed amount of data (Sun et al., 2017) while recent progress relies on additional data (Touvron et al., 2019).

## 2.3 ALGORITHMIC PROGRESS AND INCREASED EFFICIENCY

As increased sample usage might make progress in DRL look faster, increased compute usage might do so for machine learning. Grace (2013) found that hardware improvements account for half of the progress for several problems in the vicinity of AI research like computer chess, computer go and physics simulations. More recently, Fichte et al. (2020) observed the same for progress in SAT solvers. In the realm of deep learning, Hernandez & Brown (2020) investigated improvements in algorithmic efficiency by holding final performance on various benchmarks constant and analyzing how the number of FLOPS needed to reach that level changed over time. They find that this amount of compute needed to reach the performance of AlexNet (Krizhevsky et al., 2012) on ImageNet went down with a halving time of 16 months over 6 years. Looking at DawnBench (Coleman et al., 2017) submissions, they also find a cost reduction in terms of dollars of 184x[2] between October 2017 and September 2018 for reaching a fixed performance level on ImageNet. Another submission managed to reduce the training time by a factor of 808x from 10 days and 4 hours to 18 minutes in the same time interval while still cutting the cost by a factor of over nine.

## 3 METHODS

### 3.1 WE MEASURE GROWTH IN SAMPLE EFFICIENCY BY HOLDING PERFORMANCE CONSTANT

As proposed by Dulac-Arnold et al. (2019) and similar to work by Hernandez & Brown (2020) on measuring algorithmic progress, we measure progress in sample efficiency by comparing the number of samples needed for systems to reach a fixed performance level over time and define the sample efficiency of an algorithm for a given task and performance level as 1/S, where S is the number of samples needed to train to the given performance level on the task:

$$\text{Sample Efficiency}(Task, Score, Algorithm) = \frac{1}{\text{Samples } Algorithm \text{ needs for } Score \text{ on } Task}$$

Compared to the dual approach of comparing the performance of different algorithms for a fixed amount of samples, this approach has the advantage of being more comparable between different benchmarks and a lot easier to interpret: The marginal difficulty of gaining score at a given performance level could vary a lot between games, and it is unclear whether twice as much score corresponds to twice better performance in any meaningful sense (Hernandez-Orallo, 2020), while

---

[2]This might be 88x, as there is a second initial contribution which cost $1113 rather than $2323.

twice as many samples to reach the same level of performance can straightforwardly be interpreted as half the sample efficiency.

## 3.2 WE CONSIDER BOTH ATARI AND CONTINUOUS CONTROL

We look at sample efficiency in three areas: 1) for the median human-normalized score on 57 games in the Arcade Learning Environment (ALE) (Bellemare et al., 2013) using the no-op evaluation protocol and pixel observations; 2) single task performance on continuous control tasks from the rllab-benchmark (Duan et al., 2016) using simulation states[3] as input; and 3) on tasks from the DeepMind Control Suite (Tassa et al., 2018) using pixel observations as input. We chose these tasks because of their popularity and the existence of some de facto standards on evaluation and reporting protocols, which yields more mutually comparable published results. The tasks' popularity also implies attention from the research community, which ensures that they are somewhat representative of overall progress in DRL. Another advantage is the tasks' moderate difficulty and the wide range of partial solutions between failing the task and solving it optimally. This allows for a meaningful comparison of a wider set of different algorithms as more algorithms reach the performance threshold, while there is still space for improvement using advanced techniques.

## 3.3 WE TREAT EACH PUBLICATION AS A MEASUREMENT OF THE STATE-OF-THE-ART.

For our main results on progress in sample efficiency, we treat every considered result as a measurement of the state-of-the-art at publication: For each publication, we plot the sample usage (grey dot in figures 2 (a) and 3) and the state-of-the-art in sample efficiency at the time (blue stars in figures 2 (a) and 3) such that results that improve on the state-of-the-art are represented by a grey dot with a blue star in the middle. Then we fit the blue stars. Compared to fitting a dataset only containing jumps in the SOTA (grey dots with a blue star), this dampens the effect single outliers have on the fitted model and means that prolonged periods of stagnation are taken into account adequately. However, our way of fitting the SOTA is sensitive to the precise dates associated with results that did not improve on the SOTA. Another approach would be to graph the SOTA on a continuous-time axis and approximate the corresponding step function by an exponential. This solves the sensitivity to measurements that don't affect the SOTA but comes with two drawbacks: it requires an arbitrary choice of end date that affects the estimated doubling time and leads to high sensitivity to isolated early measurements. Full results for the estimated doubling times for the different approaches can be found in appendix D. Compared to the between task/score variation, the effect of the approach is usually small, but we did observe some larger effects when there have only been few and early changes to the SOTA. In these cases, the models using only jumps predict fast doubling times despite long periods of stagnation, while both other models are more conservative. On the other hand, the continuous-time model yielded the fastest doubling times for most pixel-based MuJoCo tasks, as it puts a lot of weight on improvements relative to the early, weak D4PG (Barth-Maron et al., 2018) baseline.

## 3.4 OUR STUDY IS BASED ON PREVIOUSLY PUBLISHED TRAINING CURVES

Due to the difficulty and overhead associated with the independent replication of results in DRL (Islam et al., 2017; Henderson et al., 2018), we decided to base our study on training curves in published papers. To that extent, we conducted a systematic search for papers tackling the respective benchmarks and included all relevant[4] identified papers which reported on the metrics under consideration in a sufficiently precise way. A more detailed description of the results we included and the search process we employed to identify them can be found in appendix B. While our approach implies multiple limitations of our study that are discussed in appendix C, such as a biased selection of results and the inability to tune hyperparameters, we believe that the limited methodology still offers substantial value in exploring trends in DRL sample efficiency.

---

[3]As the dynamics depend on the simulation state, training on the state is easier as seen in Yarats et al. (2019).

[4]A single paper that reported results (Doan et al. (2020)) was excluded for lack of relevance.

### 3.5 WE ALSO LOOK AT OTHER METRICS FOR ATARI

In addition to measuring progress in sample efficiency, we analyze trends in the number of samples used for overall training of agents and compare progress in state-of-the-art performance given restricted and unrestricted amounts of data to better understand the role using large amounts of samples plays for benchmark performance. As Martinez-Plumed et al. (2018) did for compute usage and task performance, we also look at shifts over time in the Pareto front that visualizes the tradeoff between sample usage and median task performance on the ALE.

## 4 RESULTS

### 4.1 THE AMOUNT OF SAMPLES USED TO TRAIN DRL AGENTS ON THE ALE AND THE SPEED AT WHICH SAMPLES ARE GENERATED HAS INCREASED RAPIDLY

Since DQN was first trained on the majority of the now standard 57 Atari games in 2015 (Mnih et al., 2015), the amount of samples per game used by the most ambitious projects to train their agents on the ALE has increased by a factor of 450 from 200 million to 90 billion as shown in figure 1 (a). This corresponds to a doubling time in sample use of around 7 months. Converted into real game time, it represents a jump from 38.6 hours (per game) to 47.6 years which was enabled by the fast speed of the simulators and running large amounts of simulations in parallel to reduce the wall-clock time needed for processing that many frames. In fact, the trend in wall-clock training time is actually reversed as can be seen in table 1: while DQN was trained for a total of 9.5 days, MuZero took only 12 hours of training to process 20 billion frames (Schrittwieser et al., 2019), which represents a speedup in utilized frames per second of 1900 in less than five years. The speedup of Ape-x and R2D2 compared to DQN can be explained by parallelization and the use of hundreds of CPU cores in addition to a single GPU. Meanwhile the faster MuZero took advantage of tens of modern TPUs.

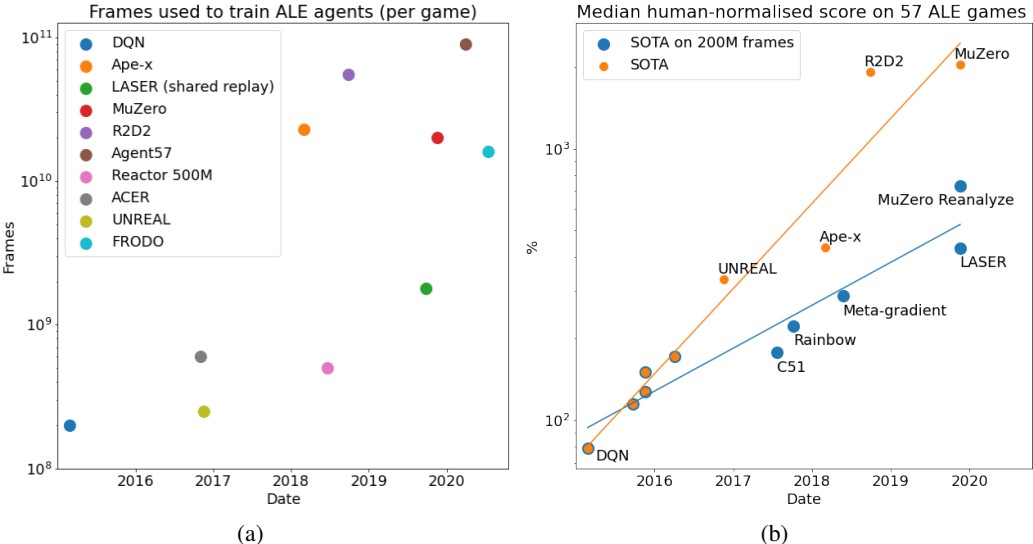

(a)                                      (b)

Figure 1: (a): Amount of frames per game used for results on Atari over time plotted on a log scale. To make the plot more readable, all work after DQN that used exactly 200 Million frames is excluded. (b): Median human-normalized score on 57 Atari games over time plotted on a log scale. Orange indicates the state-of-the-art at the time, while the state-of-the-art using 200 million frames is plotted in blue. Orange dots with blue border indicate overall state-of-the-art results that were achieved using only 200 million frames. Note that we only plot improvements in the SOTA. The corresponding estimated doubling times for jumps in the SOTA are at 12 months on an unrestricted amount of frames and at 23 months with frames restricted to 200 million per game.

| Algorithm | Date | Frames | Duration | Speed Factor |
|-----------|------|--------|----------|--------------|
| DQN (Mnih et al., 2015) | February 2015 | 200 M | 9.5 days | 1 |
| Ape-x (Horgan et al., 2018) | March 2018 | 22.8 B | 5 days | 217 |
| R2D2 (Kapturowski et al., 2018) | September 2018 | 37.5 B | 5 days | 356 |
| MuZero (Schrittwieser et al., 2019) | Nov 2019 | 20 B | 12 hours | 1900 |

Table 1: Utilized frames and the amount of wall-clock time used for training on Atari over the years. Speed factor is the speedup in the amount of frames used per second compared to DQN. The data points were chosen based on data availability and a subjective assessment of relevance.

## 4.2 PERFORMANCE ON THE ALE SEEMS TO GROW EXPONENTIALLY WITH A FIXED AMOUNT OF SAMPLES BUT TWICE AS FAST WHEN SAMPLES ARE LESS RESTRICTED

Training DQN on billions of frames would take prohibitively long (Table 1) and judging from the training curve[5], it seems unclear whether this would improve performance at all. This indicates that progress the performance on the ALE has not exclusively been driven by the massively increased amount of training frames. In fact, we observe two different trends when looking at the state-of-the-art median human-normalized score without restricting the number of frames used and the state-of-the-art performance on 200 million training frames (Figure 1 (b)). While the exact slopes of the fitted trendlines are fairly uncertain due to the limited amount of data points, especially for the unrestricted benchmark, it seems like progress on the unrestricted benchmarks is around twice as fast. This can be interpreted as roughly half of progress coming from increased sample use, while the other half comes from a combination of algorithmic improvements and more compute usage[6].

## 4.3 PROGRESS IN SAMPLE EFFICIENCY ON THE ALE CAN BE FIT BY EXPONENTIAL CURVES WITH DOUBLING TIMES OF 10 TO 18 MONTHS FOR DIFFERENT PERFORMANCE LEVELS

The improvements on 200 million frames indicate improved sample efficiency. Indeed, this is true: Figure 2 (a) shows the number of frames needed to reach the same median human-normalized score as DQN as grey dots and the state-of-the-art at the respective publication dates as blue stars. The black line shows the best exponential fit[7] for the blue dots and corresponds to a doubling time in sample efficiency of 11 months. The doubling time estimate does depend on the baseline we compare performance to and is a bit higher when later and stronger results are used (Table 2).

| Performance Baseline | DQN | DDQN | Dueling DQN | C51 |
|----------------------|-----|------|-------------|-----|
| Doubling time | 11 months | 10 months | 15 months | 18 months |

Table 2: Estimated doubling times in sample efficiency based on various performance baselines.

As LASER, the last benchmark we consider is atypical in that its training curve is unusually linear, such that its strong performance at 200 million frames does not translate to better sample efficiency and as we were not able to obtain training curves for MuZero which achieved a 731% median human-normalized score after 200 million frames in 2019, we might underestimate the actual doubling time. On the other hand, we did not find improvements for the full 57 game benchmark in 2020 and the current SOTA for sample efficiency was achieved in 2018, which points in the opposite direction. Either way, progress in DRL sample efficiency on the ALE has likely been faster than Moore's law.

## 4.4 PROGRESS ON THE ALE IS NONUNIFORM AND STAGNATION CAN OCCUR

Temporary stagnation in progress on sample efficiency happened before. For example, there have been only marginal improvements from C51 over Dueling DDQN despite a more than one and a

---

[5]Taken from the Rainbow paper (Hessel et al. 2018).

[6]In the form of larger neural networks or reusing samples for multiple training passes.

[7]As the y-axis is logarithmic.

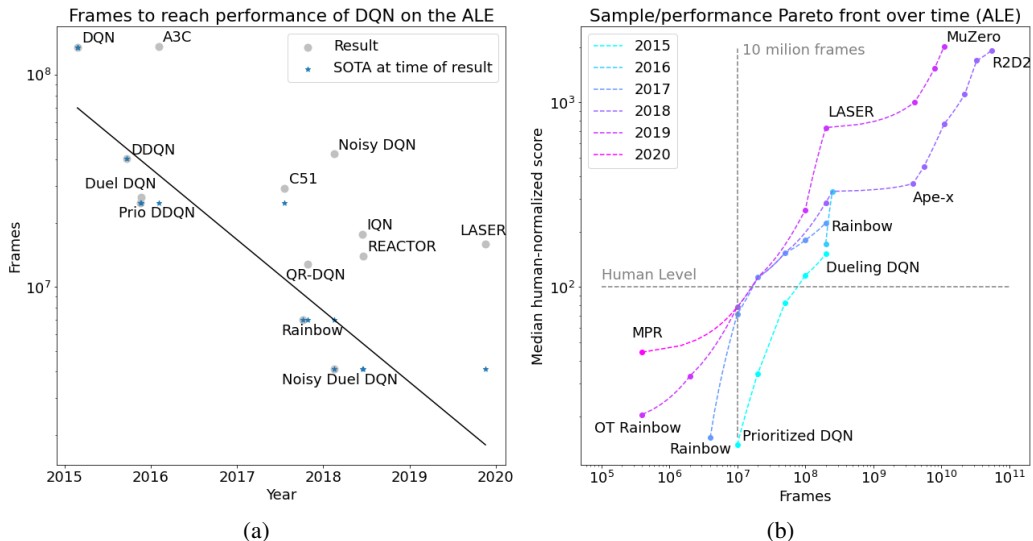

Figure 2: (a): Amount of frames needed per game to reach the same median human-normalized score as DQN over 57 games in the Arcade Learning Environment (ALE) (Bellemare et al., 2013). Grey dots indicate measurements and blue dots indicate the SOTA in sample efficiency at the time of a measurement. The linear fit on the log scale plot for the SOTA (blue dots) indicates exponential progress in sample efficiency. It corresponds to a doubling time of 11 months. (b): Pareto front concerning training frames per game and the median-normalized score on Atari on a doubly logarithmic scale. The dotted lines indicate an interpolation from the data points[8]. Results for less than 10 million frames consider 26 rather than all 57 games.

half year period between their publication dates. This lack of progress in 2016 can clearly be seen on the frames/median human-normalized score Pareto front (Figure 3 (b)). A possible explanation for the lack of substantial improvement in 2016 could be that it took some time for a breakthrough to unlock a new set of low-hanging fruit to be picked after most had been harvested in 2015. Indeed, the invention of distributional methods which estimate a distribution of returns in 2017 was rapidly followed by improvements of the approach (Dabney et al., 2018a) and combinations with other methods (Hessel et al., 2018) and enabled a lot of progress. Interestingly, distributional methods did not seem to play a large role after mid-2018, when methods discarded the distributional approach (Badia et al., 2020) or entirely moved away from approaches based on Q-learning (Schmitt et al., 2019; Schrittwieser et al., 2019). Unlike distributional approaches, distributed approaches that use multiple actors collecting experience in parallel and are thus able to generate a lot more samples started to gain importance in 2018 (Horgan et al., 2018; Kapturowski et al., 2018). This can be seen in the expansion of the Pareto front towards the right of figure 3 (b) in 2018, and is a trend that continued in 2019. But 2019 also saw increased attention to the left side of the Pareto front, the low sample regime. Interestingly, the two leftmost points on the Pareto front in 2019 representing a lot of progress essentially use Rainbow DQN (Hessel et al., 2018) which was first tested in 2017 with differently tuned hyperparameters (Kielak, 2020). This suggests that they are more indicative of the renewed attention to training with a severely restricted amount of samples, rather than of conceptual progress in sample efficient learning. While the only visible progress on the Pareto front in 2020 so far has been in the low-sample regime, this does not mean that progress in reinforcement learning has come to a halt. Rather, recent breakthroughs have been focused on reaching above 100% human-normalized score on all 57 Atari games, rather than optimising the median while neglecting progress on the hardest games (Badia et al., 2020).

## 4.5 PROGRESS IN SAMPLE EFFICIENCY MIGHT BE EVEN FASTER FOR CONTINUOUS CONTROL

Figure 3 which is structured like figure 2 (a) shows the number of frames needed to reach 2000 score in the state-based HalfCheetah task from the rllab/gym benchmark and the number of frames needed

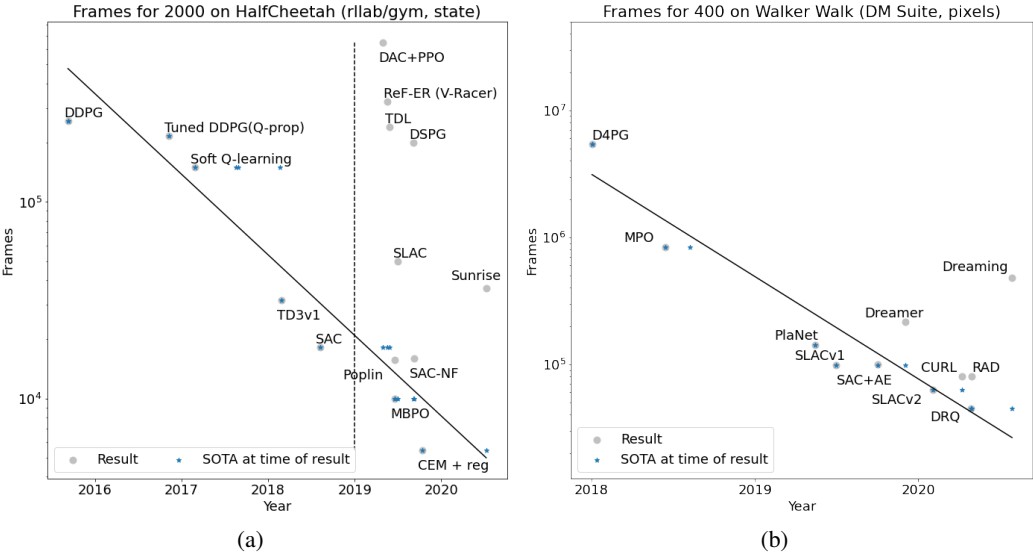

Figure 3: (a): Frames needed to reach 2000 score on the Half-Cheetah task from OpenAI gym over time. Grey dots indicate results and blue dots indicate the SOTA in sample efficiency at the time of a result. The black line is the best fitting exponential model for the SOTA (blue dots). It corresponds to a doubling time in sample efficiency of 9 months. The vertical line indicates the switch from results on gym-v1 to v2 and ambiguous versions of gym. (b): Frames needed to reach 400 score in on the Walker Walk task from the DeepMind Control Suite over time. Grey dots indicate results and blue dots indicate the SOTA at the time of a result. The black line corresponds to a doubling time in sample efficiency of 5 months. Note the different scaling of the y-axes in (a) and (b).

to reach 400 score on the pixel-based Walker Walk task from the DeepMind Control Suite over time. The SOTA is fitted well by an exponential model with a doubling time of 9 months for HalfCheetah and 5 months for Walker walk. Averaged over all task/score combinations we considered[9], the average doubling time was 11 months for state-based continuous control and 5 months for the pixel-based case, the latter of which is faster than the trend in overall sample use on the ALE. The amount of included results rose in 2019 for both types of tasks[10], but this does not seem to correspond to faster progress in sample efficiency. As the Control Suite was only established in 2018, the overall time frame we considered was shorter for the pixel-based tasks, such that the low doubling time is less robust than our other results. Another reason to be careful with extrapolating from these very short doubling times is that a naive extrapolation would predict that 400 score on Walker Walk will be reachable using a single sample within six and a half years. As learning consistent behaviour from a single sample is most likely impossible, the exponential model has to break down rather soon, at least for tasks that are already solvable with relatively few samples.

## 5 MAIN CONTRIBUTORS TO IMPROVED SAMPLE EFFICIENCY

### 5.1 IMPROVED OFF-POLICY LEARNING

In all three domains we considered, almost all improvements in sample efficiency were achieved by off-policy algorithms that can use the same sample for multiple updates and allow for more elaborate exploration strategies. While we won't enumerate all of these improvements, two general strategies stand out and played a role in multiple domains: Counteracting maximization bias by employing a second Q-network (van Hasselt et al., 2016; Fujimoto et al., 2018), and improved exploration by using parametric noise (Fortunato et al., 2017) or by encouraging high entropy policies (Haarnoja et al., 2017; 2018).

---

[9]Detailed results for all task/score combinations can be found in Appendix D.

[10]For gym/rllab, this is in part due to the inclusion of results from ambiguous gym versions after 2018.

## 5.2 Model-based RL

While not yet up to par with DQN performance on the ALE, model-based algorithms have played an important role for improved sample efficiency in continuous control tasks and also helped to catalyze recent improvements in the low-sample regime on the ALE (Kaiser et al., 2019). Salient features of some model-based approaches with strong sample efficiency include splitting the transition model into a deterministic and a stochastic part and using latent overshooting, which incentivizes longer model rollouts to be accurate (Hafner et al., 2019a), as well as preventing model trajectories from diverging too far from the training distribution by employing a penalty term (Boney et al., 2020) or by using short rollouts starting in states from a replay buffer (Janner et al., 2019).

## 5.3 Auxiliary objectives

As sample efficiency is a lot better for state-based continuous control than for the pixel-based version on the DeepMind Control Suite, various recent approaches tackling the pixel-based version employed auxiliary objectives to quickly learn a state representation that is amenable to sample efficient learning. Approaches with strong sample efficiency learnt an autoencoder on frames (Yarats et al., 2019) or a low-dimensional latent dynamics model (Lee et al., 2019) for representation learning, or incentivized Q-values (Kostrikov et al., 2020) or latent representations (Srinivas et al., 2020) to be similar for different image augmentations applied to the same game frame .

## 5.4 Incentives for sample efficiency can have a large effect

Often, explicitly tuning hyperparameters for sample efficiency can have a large effect as can be seen in the substantial improvements to Rainbow's performance in the low-data regime where sample efficiency for median performance on 27 Atari games improved by more than 10x when hyperparameters were tuned accordingly (van Hasselt et al., 2019; Kielak, 2020). This was achieved by increasing the frequency of updates to the Q network, relative to the frequency of sample collection, such that comparable gains seem plausible for other approaches using experience replay. Furthermore, incentives for sample efficiency also influence the focus on sample efficiency in algorithm development. This might in part explain the comparatively fast doubling times for the pixel-based continuous control domain where many recent results (Hafner et al., 2019b; Lee et al., 2019) explicitly focus on sample efficiency.

## 5.5 The role of compute

A lot of the previously discussed improvements like updating the Q-network more often or using a second Q-network do require additional compute. However, these additional requirements do not seem to keep up with the 3.4 month doubling times in compute usage for the largest AI projects, as they are usually way smaller than the increase of a factor of 32x per sample reported in Kielak (2020). While experiments suggest that increases in network size which exacerbate the compute requirement per network update can lead to improved DRL performance (Espeholt et al., 2018), network size stayed constant for the different versions of DQN that dominate our comparison of sample efficiency on the ALE and improved sample efficiency did not seem to strongly correspond to larger network size, more generally. Nevertheless, further research into the role of compute for RL performance and sample efficiency would certainly be of value.

## 6 Discussion

Together with cost reductions in robotics hardware (Yang et al., 2019a; Rahme, 2020; Ahn et al., 2020) sustained progress in sample efficiency might enable ample applications of DRL in the real world. In addition, we might see compute-powered improvements in simulation quality, more unsupervised pretraining for representation learning on large task-adjacent datasets and further progress in transferring learnt behaviour from simulations to reality (Peng et al., 2018; James et al., 2019; van Baar et al., 2019). These factors might reduce the cost of real world DRL by providing a reasonable starting policy that makes human intervention to reset the learning system (Zhu et al., 2020) or prevent catastrophes less important. Such a baseline policy could also be obtained from imitation learning if we are able to create systems that robustly interact with the real world by means other

than DRL. Once economically relevant applications of DRL become more widespread, sample efficiency is likely to gain economic relevance which would fuel further progress in sample efficiency. However, widespread structurally similar real world applications of DRL could eventually alleviate the need for better sample efficiency, either by providing immense datasets that can be used to leverage progress in offline DRL (Agarwal et al., 2020) or when combined with improved generalization capabilities (Goyal et al., 2019; Sekar et al., 2020).

We want to stress that there are several limitations to our study: the reliance on published training curves can add noise or even bias, as a single noisy outlier can greatly influence our estimate for the SOTA. We were also unable to tune hyperparameters for sample efficiency which means that we cannot clearly separate more cleverly tuned hyperparameters from more substantial progress in sample efficiency. Furthermore, we focus on only three types of tasks and neither consider progress in robotics or offline DRL sample efficiency, nor general progress in sim2real and transfer learning, all of which might play an important role for real world application of DRL. In particular, we believe that progress in offline DRL might be slower, as improved exploration strategies cannot be leveraged for sample efficiency in that case. Lastly, it could be argued that unlocking new capabilities, a dimension we ignored, corresponds to more progress in sample efficiency than getting to previously unlocked capabilities slightly faster. A more detailed discussion of these limitations can be found in appendix C. Furthermore. there are many other ways in which extrapolating the presented trends could be misleading: for example, progress might speed up a lot once the first commercial application of real world DRL create stronger economic incentives for sample efficiency or progress might slow down significantly as we come closer to fundamental limits on sample efficiency. Progress might also slow down significantly, if algorithmic progress played a smaller role in recent progress than the explicit tuning of hyperparameters for sample efficiency. While tuning hyperparameters can lead to a lot of short term gains, there are likely diminishing returns and unlike insights, hyperparameters are often not transferable to novel settings. It is also worth keeping in mind, that naive extrapolation of the trends we observed would lead to instantaneous learning of strong behaviour on some of the tasks we observed, within the decade, which seems highly implausible.

## 7    CONCLUSION

We find considerable progress in the sample efficiency of DRL at rates comparable to progress in algorithmic efficiency in deep learning. If the trends we observed proved to be robust and continued, the huge amounts of simulated data that are currently necessary to achieve state-of-the-art results in DRL might not be required for future applications such that training in real world contexts could become feasible. However, this should be taken with a grain of salt as relying on the extrapolation of trends that have only been observed for a small time frame using noisy data can be problematic.

A reiteration of our study that reproduces results rather than relying on training curves and tunes hyperparameters for sample efficiency would fix many of our study's limitations, albeit at a substantial time investment, given the difficulty of reproducing results in DRL, the large number of publications we considered, and the lack of verified open-source implementations of many SOTA algorithms. Still, such a study could focus on a subset of results that are easily reproducible or performed well in our study. Simpler investigations into how sample efficiency depends on hyperparameter tuning for different algorithms could already help to clarify the role of tuning incentives.

Similarly, further monitoring of the observed trends' continuity, as well as research on the drivers of the observed progress and differences between various domains would be of value. Researchers in DRL can support this effort by reporting the number of samples that were needed by their final algorithm to reach salient performance levels. In particular, sample efficiency should be reported for performance levels that were used in earlier publication to make the quantification of trends more easy.

## ACKNOWLEDGMENTS

Hidden for anonymity. Will be added to camera-ready version

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

## A BENCHMARK DESCRIPTIONS

### A.1 ATARI (ALE)

The Arcade Learning Environment (Bellemare et al., 2013) provides an interface to a variety of Atari 2600 games that is adapted to the needs of RL research. When run in real time, the simulator generates 60 frames per second, but more than 6000 frames per second can be created by a single simulator. Each frame consists of 210x160 pixels on a 128-colour palette, but preprocessing often involves downscaling to 84x84 pixels and conversion to grayscale images (Mnih et al., 2015). While some authors use prespecified features of these pixels as the input to their algorithm (Liang et al., 2015), we only consider methods that directly learn from pixel input. Many of the games cannot be modelled as a Markov Decision Process (Sutton & Barto, 2018), as vital information is often not visible in every frame[11]. To circumvent this issue, it is common practice to stack multiple frames

---

[11]For example, enemy projectiles in the game Space Invaders can sometimes be invisible for multiple frames.

and use them as a single observation (Mnih et al., 2015). Still, this does not always yield full observability (Hausknecht & Stone, 2015), such that many of the more successful solution methods employ recurrent neural networks (Schmitt et al., 2019; Kapturowski et al., 2018). The action space of the environments in the ALE is discrete and consists of up to 18 actions representing different inputs on the original system's joystick controller. The initial DRL benchmark set by DQN (Mnih et al., 2013) consisted of seven games in the ALE, but the nature version of the DQN paper reported 49 games in 2015 (Mnih et al., 2015). Eight additional games were added in the Double DQN paper (van Hasselt et al., 2016) and the resulting 57 games have been the standard suite of games that ambitious projects try to tackle, since. To make scores comparable between games, they are often reported as human-normalized scores: professional human game testers repeatedly played each game after around 2 hours of practice and the mean score was collected. Then, the agent's results are normalized for this mean score to be at 100%, while the average score for random play is at 0%[12] (Mnih et al., 2015). As the ALE is deterministic, evaluation is often done using a random amount of up to 30 no-op actions at the beginning of each episode to avoid assigning high scores to algorithms that simply overfit to deterministic trajectories (Mnih et al., 2015). We focus on the median human-normalized score for multiple reasons: First, historically results and especially training curves are published more commonly for this metric than for others. Apart from that, we care about overall progress, such that an aggregate measure is preferable to single-game scores. Lastly, the mean score is often dominated by a few outlier games and a less robust indicator of improvements. Still, the median has its own problems: some games are harder to solve than others and median performance is not sensitive to progress on these games.

## A.2 CONTINUOUS CONTROL IN MUJOCO

MuJoCo (Todorov et al., 2012) which stands for "Multi-Joint dynamics with Contact" is a physics engine originally built for model-based control. It enables users to combine bodies with tendons, different types joints and controllable actuators in a fast and parallelizable simulation[13] that can easily be visualized. In 2015, MuJoCo was used in early projects working on DRL for continuous control with a focus on locomotion tasks where rewards are obtained by making a simple humanoid or other "robot" move (Heess et al., 2015; Lillicrap et al., 2015; Schulman et al., 2015). In the next year, Duan et al. (2016) proposed the rllab benchmark to allow for better quantification of progress in continuous control. The benchmark includes adaptions of six locomotion tasks of varying dimensionality and a set of hierarchical tasks that combine locomotion with more abstract higher-level goals like solving a maze. Adaptions of many of the rllab environments are included in OpenAI gym[14] (Brockman et al., 2016), a popular toolkit that provides a common interface for a variety of RL benchmarks. However, even though the documentation[15] frames gym as an attempt to standardize environments used in publications as subtle differences can have big impacts in RL, it is possible that it had the opposite effect: there are multiple versions of the gym environments that might behave differently, even if a baseline algorithm shows similar performance for most but not all environments when trained on different versions[16] and many publications do not specify which of these versions they use (Nachum et al., 2017; Clavera et al., 2018). Furthermore, at least the "Humanoid" task clearly behaves differently in the rllab implementation and the gym version (Haarnoja et al., 2018). Scores for different rllab/gym MuJoCo tasks can be on different scales and normalization is rarely used which makes measuring aggregate performance on the benchmark hard to measure. The incomparability of scores between tasks in the rllab benchmark was one of the motivations behind the introduction of the DeepMind Control suite (Tassa et al., 2018), another interface to MuJoCo in which tasks similar to the ones in rllab are combined with binary rewards depending on proximity to a goal state and evaluation over 1000 steps, such that attainable overall scores always fall in the interval [0,1000]. Despite the comparability of scores, most publications we looked at did not actually provide training curves for the overall average or median performance on the DeepMind control suite.

For rllab/gym, we looked at Walker2d, Humanoid, HalfCheetah, Hopper and Ant where an armless humanoid, a humanoid with arms, a two-legged "cheetah", a monopod or a quadruped have to move

---

[12]Human world-records have a median human-normalized score of 4400% (Toromanoff et al., 2019).

[13]The original publication is around 5000 times faster than real time on a machine with 12 cores.

[14]https://github.com/openai/gym

[15]http://gym.openai.com/docs/

[16]https://github.com/openai/gym/pull/834

forward. For the Deep Mind Control Suite, we looked at Walker Walk, Cheetah Run, Finger Spin and Ball in Cup Catch. The first two are analogous to Walker2D and HalfCheetah, while in Finger Spin a finger has to rotate a hinged body and Ball in Cup Catch resembles the Cup-and-ball children's toy.

## B DATA COLLECTION

The publications we considered in our results were identified in three ways: 1) drawing from our existing domain knowledge, 2) by searching the top 30 entries in google scholar sorted by relevance (over all time, as well as since 2019 and since 2020) that cite either the paper introducing the environment, the particular benchmark, or the first major application of DRL to the particular benchmark for results in a comparable setting and 3) by adding all publications for which quantitative results in a comparable setting were cited in one of the publications already on our list and repeating this procedure until no new publications were identified. For papers with multiple versions, the publishing date of the first version that contained the relevant result is used as the date in all of our plots[17]. In case the training curve was not taken from the original paper, we considered the date of the first preprint of the original paper that contained the same reported main results as the most recent version[18] of the original paper at the publication time of the paper the data was taken from. We report the number of samples after which the training curve first reaches the specified score level. In general, we report the current state-of-the-art for a particular task/score combination at the publication date of each considered paper which was published after the first publication reaching that performance level on that task. We did not take the resolution at which training curves were generated into account such that some of our measured values are actually interpolated.

### B.1 ATARI

The reported results for Atari consist of the best trained agent's results using 30 no-op actions in the beginning for every game. We considered the median human-normalized score for DQN (Mnih et al., 2015), Double DQN (van Hasselt et al., 2016), A3C (Mnih et al., 2016), Dueling DQN (Wang et al., 2016a), Prioritized double DQN and Prioritized dueling DQN (Schaul et al., 2015), C51 (Bellemare et al., 2017), Rainbow (Hessel et al., 2018), IMPALA (Espeholt et al., 2018), QR-DQN (Dabney et al., 2018a), Noisy Dueling DQN (Fortunato et al., 2017), Ape-x (Horgan et al., 2018), Meta-gradient (Xu et al., 2018), IQN (Dabney et al., 2018b), REACTOR 500M (Gruslys et al., 2017), R2D2 (Kapturowski et al., 2018), C51-IDS (Nikolov et al., 2018), LASER (Schmitt et al., 2019), FQF (Yang et al., 2019b), MuZero and MuZero Reanalyze (Schrittwieser et al., 2019), as well as Agent57 (Badia et al., 2020) for figure 1 but excluded results among these that used exactly 200 million frames in figure 1 (a) and only included state-of-the-art performance (on 200 million or an unrestricted amount of frames) in figure 1 (b). Any of these results for which we found sufficiently detailed training curves for the median human-normalized score in the paper or another publication were included in figure 2 (a). These were DQN, DDQN, Prioritized DDQN, Dueling DDQN, C51, Rainbow, Noisy DQN and Noisy Dueling DQN, QR-DQN, IQN, LASER, REACTOR and A3C. Of the results that report median human-normalized performance and the amount of frames used for training we initially identified, PGQ (O'Donoghue et al., 2016), FRODO (Xu et al., 2020) and C-Trace DQN (Rowland et al., 2020) were excluded from figure 1 (b) and 2 (a) and A1 due to a lack of clarity about the use of no-op starts but are still used in figure 1 (b). ACER (Wang et al., 2016b) is also not included in figure 2 (a) and A1 due to unclearly marked axes and missing clarity about what a step consists of in their training curve plots. The performance under the no-op protocol for UNREAL is taken from the Ape-x paper.

C51-IDS was only evaluated on 55 games, not on all 57. As this does not strongly affect the median, the results are still included. Most results were obtained using a single set of hyperparameters for all games, but Ape-x and UNREAL used separately tuned hyperparameters for every game. This does not affect the doubling times in figure 2 (a), as UNREAL was excluded because of its evaluation protocol and the Ape-x publication did not provide training curves. The results reported for MuZero and R2D2 in the respective publications differ from the ones reported in the Agent57 paper and we use the ones from the original publication. The same goes for the number of samples reported for

---

[17]Usually, this would be an arxiv preprint.

[18]As of the 10th of august 2020.

R2D2, which are reported to be lower in the MuZero publication than in the original one. We used the training curves provided in the Rainbow paper for A3C, DQN, DDQN, Dueling DQN, Noisy DQN, C51 and Rainbow and the training curves from the respective paper for Noisy Dueling DQN, QR-DQN, IQN, REACTOR and LASER to compare algorithm performance.

The performance results in figure 2 (b) consider the same publications as the previous figures, as well as Simple (Kaiser et al., 2019), Data-efficient Rainbow (van Hasselt et al., 2019), OTRainbow (Kielak, 2020), Efficient DQN+DrQ (Kostrikov et al., 2020) and MPR (Schwarzer et al., 2020). Some of the reported results are approximately lower bounds for the true score, as they are derived from training curves and might include non-greedy actions in the evaluation run. We essentially made an arbitrary choice about the amounts of frames for which we measured performance on the training curves. Whenever results from training curves and the final reported performance for the same amount of used frames conflicted, we chose the result with the larger score, which was usually the reported final result. The results using less than 10 million frames come from Kielak (2020) and Kostrikov et al. (2020) and are medians over a subset of 26 instead of the full 57 games and it is not clear, whether or not no-op actions were used for these results. In particular, the results for 2017 in that regime are from runs of a slightly simplified version of Rainbow with the original hyperparameters in 2019 (Kielak, 2020). The median human-normalized scores for Simple (Kaiser et al., 2019) are taken from the MPR paper.

The training durations used in Table 1 are taken from the Ape-x and Muzero papers.

### B.2 MuJoCo-gym

Due to the variety of results we considered for continuous control, the protocols for hyperparameter tuning can vary considerably between papers. Most publications don't tune hyperparameters on a per-task basis[19] which can make results from publications that used a larger variety of tasks seem worse than they would be with more fine-grained parameter tuning.

For the MuJoCo-gym environments, we only consider results that were reported with the v1 version of the respective environment up to 2019 as the earliest publication of the latest result we found for v1 (Abdolmaleki et al., 2018a) came out in December 2018, but include results that use v2 or an ambiguous version from 2019 and 2020. Over all, we considered TRPO (Schulman et al., 2015), DDPG (Lillicrap et al., 2015), Q-Prop (Gu et al., 2016), Soft Q-learning (Haarnoja et al., 2017), ACKTR (Wu et al., 2017), PPO (Schulman et al., 2017), Clipped Action Policy Gradients (Fujita & Maeda, 2018), TD3 (Fujimoto et al., 2018), STEVE (Buckman et al., 2018), SAC (Haarnoja et al., 2018) and Relative Entropy Regularized Policy Iteration (Abdolmaleki et al., 2018a) for gym-v1. Apart from that we considered Target Distribution Learning (Zhang et al., 2020), MBPO (Janner et al., 2019), POPLIN (Wang & Ba, 2019), SLAC (Lee et al., 2019), DSPG (Shi et al., 2019), TD3 with CrossRenorm (Bhatt et al., 2019), SAC-NF (Mazoure et al., 2020), Regularized CEM (Boney et al., 2020) , DAC (Zhang & Whiteson, 2019) and SUNRISE (Lee et al., 2020) which all either use gym-v2 or an ambiguous versions of gym. As the reported results for the Walker task in the first version of the TD3 paper are already strong but improved upon by the results in the second version, we include both versions, labelled as TD3v1 and TD3v2.

The results for Soft Q-learning are taken from the SAC paper, the results for MVE are from the STEVE paper and the results for TRPO, DDPG and ACKTR, as well as the results on Ant for PPO, are taken from the TD3 Paper (v2). We used the reported results for DDPG with the original hyperparameters rather than the improved version the authors of the TD3 paper suggest. However, we also include the tuned version of DDPG from the Q-Propr paper, as it is the first reported method that reached a score of 6000 in half cheetah-v1. We assign the publication date of Q-prop to this method. The results for PPO on the Humanoid task are from the SAC paper.

The results in regularized CEM and Q-Prop were reported in episodes rather than time steps. We converted them into time steps by multiplying by the standard episode length of 1000, but some tasks might include non-step based alternative termination conditions, such that these numbers might be an overestimate. We excluded Infobot (Goyal et al., 2019) and the results by Rajeswaran et al. (2017) in RBF-policies due to unclarity what an "iteration" on the x-axis of their learning curve constitutes. We also excluded ESAC (Suri et al., 2020) due to unclear scaling of the x-axis in their plots (for

---

[19]The action repeat parameter in the DeepMind Control Suite is an exception to this rule.

example the distance between 0 and 0.5 was clearly different from the distance from 0.5 to 1 for their Walker results). Next, Expected Policy Gradients (Ciosek & Whiteson, 2018) was excluded due to unclearly marked axes. Lastly, we excluded the variant of the (slightly dated) Reinforce algorithm described by Doan et al. (2020), as this paper had a theoretical focus and their results did not even get close to the score thresholds we looked at. We were not able to obtain any reasonable measurements for MVE, STEVE and TD3+CrossRenorm due to insufficient resolution around the zero point of the plot on half cheetah for 2000 score. As the graphs were cut off below the respective score level, we did not include scores for Relative Entropy Regularized Policy Iteration on 1000 score for Ant. Also, note that not all papers reported results for all the environments we considered.

## B.3 DEEPMIND CONTROL SUITE (ON PIXELS)

For the DeepMind Control Suite, we considered D4PG (Barth-Maron et al., 2018), SAC (Haarnoja et al., 2018), MPO (Abdolmaleki et al., 2018b), PlaNet (Hafner et al., 2019a), SLAC (Lee et al., 2019), SAC+AE (Yarats et al., 2019), Dreamer (Hafner et al., 2019b), CURL (Srinivas et al., 2020), DRQ (Kostrikov et al., 2020), RAD (Laskin et al., 2020), SUNRISE (Lee et al., 2020) and Dreaming (Okada & Taniguchi, 2020). We excluded the method described in Grill et al. (2020) due to uncertainty about the meaning of "learner step" in the reported learning curves. Also note that not all papers reported results for all the environments we considered.

As the reported results in the first version of the SLAC paper are already strong but improved upon by the results in the second version, we include both versions, labelled as SLACv1 and SLACv2. We used the training curves in the ablation study ("Figure 8") for CURL, as the axes in the main comparison are labeled unclearly. As the axes labeling of this figure has been updated after the first version of the paper without the rest of the figure changing, we use the updated axis labels for our analysis. The data for MPO comes from the SLAC paper (v2). The results for SAC come from the SAC+AE paper. The data, as well as the publication date for D4PG, were taken from the DeepMind Control Suite paper (as this included D4PG results before the D4PG paper was published). As the original publication of the PlaNet results reported the median, not the mean of training curve over different runs like most other results, we took the results from the SLAC paper (v2). Dreamer also reported results for PlaNet, but these seemed incompatible with the median results reported in the PlaNet paper: The 5% quantile for Cheetah run is reported above the final performance of D4PG (524 score) at around 600 score starting at around 1000 (1000 step) episodes in the PlaNet paper, while the mean reported in Dreamer consistently stays below the performance of D4PG for the whole 5 million step training. As the scores are bounded below by 0, the underlying results are clearly different. Next, we assumed that the x-axis for Finger Spin reported in Sunrise is scaled in thousands as for all other DeepMind Control Suites in the corresponding paper, as the results would otherwise be inconsistent with the reported final results and multiple orders of magnitude better than any previous results, which was not discussed by the authors and would be very surprising given that we don't see anything comparable for any other environment. Dreamer uses 5000 random exploration steps before beginning with policy training. We assumed, that these steps are accounted for in the training curves. Either way, as Dreamer was only briefly state-of-the art for 600 score on Cheetah 600 and no other task/score combination we looked at, and because Dreamer needed at least 200000 environment steps for reaching one of the specified score levels in any of our tasks, the effect of this assumption on our results is minimal. There is a similar problem with Dreaming, whose authors claim to use similar hyperparameters to Dreamer. While the environment steps needed to reach prespecified score combinations reached levels close to 60000 in some cases, the larger effect of our assumption on Dreaming's sample efficiency does not afffect our overall results, as Dreaming never reached state-of-the art performance for any of the score/task combinations we investigated.

As the graphs were cut off below the respective score level, we did not include scores for Sunrise for 400 score on Walker Walk and 400 as well as 600 score on Finger Spin. Again, ESAC (Suri et al., 2020) was excluded due to unclear scaling of the x-axis.

Note, that later versions of SLAC report using 10000 environment steps before training to pretrain the model. While they report that these steps are included in their training curves, it is possible that they were used, but neither reported nor included in the plots in the earlier version we considered, as they were mentioned in the DRQ paper, which was published before version 3 of the SLAC paper. The values reported in our analysis assume that the reporting in the first two versions of the SLAC paper is correct. We performed a sensitivity analysis and found that doubling times changed by at

most 0.3 months, when the steps were added to all values for SLACv1 and SLACv2, for both gym and the Control Suite.

## C  LIMITATIONS

### C.1  WE ONLY CONSIDER THREE TYPES OF RL TASKS

While we cover two major axes of variation within RL tasks by including both pixel-based and state-based tasks as well as discrete action and continuous action tasks, we observe a lot of between-task variation in sample efficiency doubling times. It is very plausible that this variation would be even bigger if other types of problem classes like hard-exploration problems (Ecoffet et al., 2019) were considered as well. Also, we chose tasks by popularity. This might introduce bias, as popularity most likely correlates with sustained speed of progress.

### C.2  THE LACK OF COMPREHENSIVE COLLECTIONS OF BENCHMARK RESULTS IMPLIES THAT WE MIGHT HAVE MISSED RELEVANT RESULTS.

While we conducted a fairly extensive literature review to identify relevant publications, we may have missed some results, as there sadly is no comprehensive repository that collects benchmark results for RL. We also had to exclude ALE results that did not report median human-normalized score from our analysis. Still, we were able to look at more than ten data points for most task/score combinations we considered. Relatedly, our literature review might have identified more recent non-SOTA results with a higher probability than older non-SOTA results as we searched for the most highly cited papers overall but also the most highly cited papers from the last one and two years to avoid missing improvements on the SOTA that did not yet have had enough time to compete with older papers' citation counts.

### C.3  RELYING ON PUBLISHED LEARNING CURVES IMPLIES NOISY RESULTS

Reporting the number of samples after which the training curve first reaches the specified score level introduces some amount of bias and noise as training performance randomly fluctuates. More generally, the reliance on training curves reported in other publications forced us to interpolate some results when learning curve plots had insufficient resolution and to exclude a few results for which there was insufficient resolution around zero frames or the plot was cut off above the relevant score level. Also, the exact evaluation protocol used for generating training curves on the ALE was often ambiguous and different publications might have uses subtly different evaluation protocols (Machado et al., 2018). In particular, we observed that the provided training curves rarely reached the reported maximum performance for Atari. As this effect is rather consistent between algorithms it cannot simply be explained by noise in the evaluation as this would lead to the reversed effect similarly often. While the effect is of similar magnitude for different algorithms it is second strongest for DQN, the earliest algorithm we considered, which might inflate perceived progress by making the baseline result worse. Lastly, the possibility of plotting or transcription errors is particularly problematic because we estimate doubling times in SOTA sample efficiency and the SOTA could be set to an extreme value for a long time by a noisy outlier which might explain the bad fit and long doubling time for C51, where the measured state-of-the-art only changes once in three years (Figure 4). Luckily, we can see the state-of-the-art changing regularly in most of our plots which indicates that this problem is not too severe.

### C.4  WE WERE NOT ABLE TO TUNE HYPERPARAMETERS

As we did not replicate any experiments, we were unable to tune different algorithms to reach the respective baseline performance as fast as possible. This is problematic because the easy generation of samples using additional compute in simulated tasks combined with the usual focus on final performance[20] reduces the incentive for sample efficiency in published results. This effect would have little consequence if it was similarly large for all publications but distorts observed trends in sample efficiency if the focus on sample efficiency increases over time, which has arguably happened. On

---

[20]As evidenced by highly cited papers like Wang et al. (2016a) reporting no training curves.

the other hand, our trendlines don't seem to systematically underestimate the sample usage of early results by much, which would be expected if the described effect was large.

## C.5 THE AMOUNT OF SAMPLES USED BY THE FINAL ALGORITHM UNDERESTIMATES THE AMOUNT OF SAMPLE USED IN RESEARCH AND DEVELOPMENT.

Similar to previous work on algorithmic efficiency in machine learning (Hernandez & Brown, 2020), our analysis only considers samples that were used for the training the final agent. However, the number of samples that are used to solve challenging DRL problems can often greatly exceed this number as exemplified by OpenAI five, for which a rerun with the final network architecture took less than a third as long as the initial training during which the network was repeatedly improved (Berner et al., 2019). More generically, hyperparameter tuning requires samples which are often not reported. Depending on how much good hyperparameters generalize between tasks, this might be less of an issue for real world applications, as they could be tuned in simulations.

## C.6 SOLVING HARD, PREVIOUSLY UNSOLVED PROBLEMS CAN BE INTERPRETED AS MORE PROGRESS IN SAMPLE EFFICIENCY THAN SOLVING OLD TASKS MORE QUICKLY

As with algorithmic efficiency in machine learning (Hernandez & Brown, 2020), one could argue that most progress in sample efficiency is actually represented in the ability to solve previously unsolved tasks, rather than solving already solved tasks using fewer samples. However, the automation of tasks that can currently be solved in simulations like autonomous driving (Sallab et al., 2016), adaptive traffic light control (Liang et al., 2018) and piloting fighter jets (Tucker, 2020) would already come with a lot of economic or military value. Ultimately, it does not matter, whether we could train a robot to manufacture an abacus or a supercomputer in a simulation, as long as we struggle to train it to reliably manipulate simple objects in the real world (Zhu et al., 2020).

## C.7 WE DO NOT MEASURE PROGRESS IN ROBOTICS, SIM2REAL OR TRANSFER LEARNING.

Arguably, progress in robotics and other real world applications of DRL is what we ultimately care about. However, while there exist benchmark tasks for robotics (Mahmood et al., 2018; Yang et al., 2019a; Ahn et al., 2020), these benchmarks have received little attention: Mahmood et al. 2018 was only cited 31 times and only one of the citing publications contained a result on the benchmarks while he other two benchmark publications were not even cited ten times. This makes quantifying progress difficult. We also do not assess progress in simulation quality, which would reduce the need for sample efficient algorithms, even if successfully calibrating these simulations is challenging (Irpan, 2019; OpenAI et al., 2020) and does require the use of real transition data as ground truth. Similarly, we don't measure progress in sim2real including dynamics randomization and related approaches where an agent is trained on a variety of differently calibrated simulations to increase its robustness to simulation parameters (Peng et al., 2018; James et al., 2019; van Baar et al., 2019) or progress in RL transfer learning and generalization (Goyal et al., 2019; Sekar et al., 2020) which might reduce the need for calibration data.

## C.8 PROGRESS MIGHT BE SLOWER FOR OFFLINE REINFORCEMENT LEARNING

In the offline RL setting (Levine et al., 2020; Agarwal et al., 2020), the agent learns from a fixed dataset rather than from its own experience. As it might be easier to collect data produced by a human operator or an inefficient but save heuristic rather than an untrained agent, offline learning could play an important role for many real world applications of RL[21] (Agarwal et al., 2020; Fu et al., 2020). Due to the fixed dataset, improved exploration strategies cannot help with progress in offline RL such that we expect progress in offline RL sample efficiency to be slower than general progress in RL sample efficiency in the long run. This means we might be overestimating progress relevant for real world applications of DRL if most of these were using the offline setting. Further investigations into trends in sample efficiency in offline RL would be valuable but challenging due to the lack of established evaluation protocols (Levine et al., 2020).

---

[21]Especially if the learnt task has to be carried out anyways such that offline samples are cheap.

## D  DETAILED RESULTS

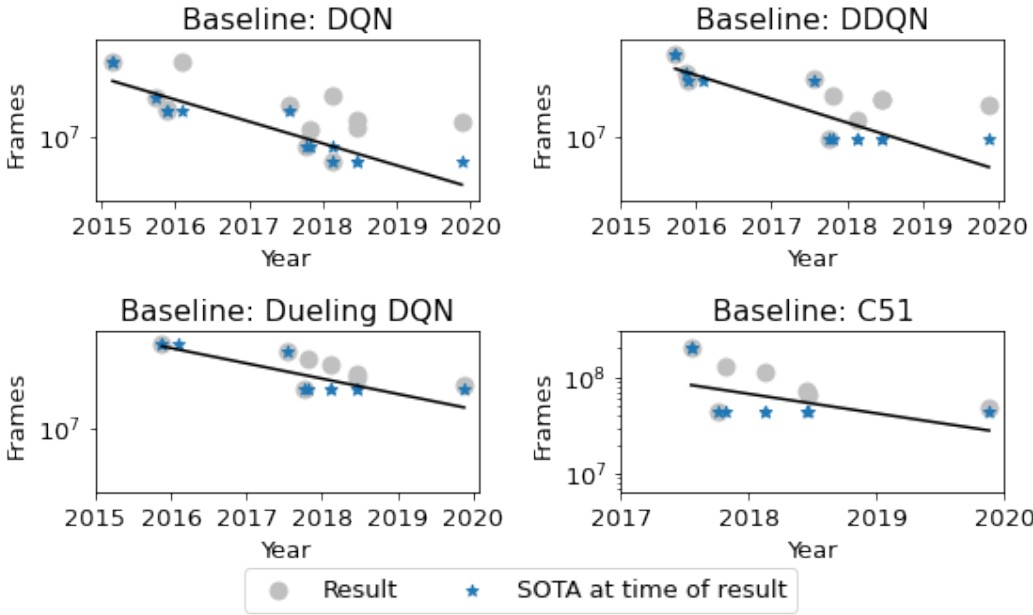

Figure 4: Frames needed to reach the same median human-normalized score as various baselines over time. Graphs are normalized such that graphs with the same slope indicate the same doubling time, despite different measurement horizons. The black line is the best fitting exponential model for the SOTA (all SOTA).

| Baseline | Doubling Time (All SOTA) | Doubling Time (Jumps only) | Doubling Time (Continuous until 18.11.2019) | Doubling Time (Continuous until 10.08.2020) |
|---|---|---|---|---|
| DQN | 10.8 months | 8.2 months | 10.4 months | 12.2 months |
| DDQN | 9.9 months | 6.5 months | 10.6 months | 12.8 months |
| Dueling DQN | 15.4 months | 14 months | 13.8 months | 17 months |
| C51 | 17.8 months | 1.2 months | 25.6 months | 42.6 months |

Table 3: Estimated doubling times in sample efficiency for median human-normalized score on the ALE. The 18th of November 2018 is the publication date of the last result that was considered for measuring sample efficiency on the ALE.

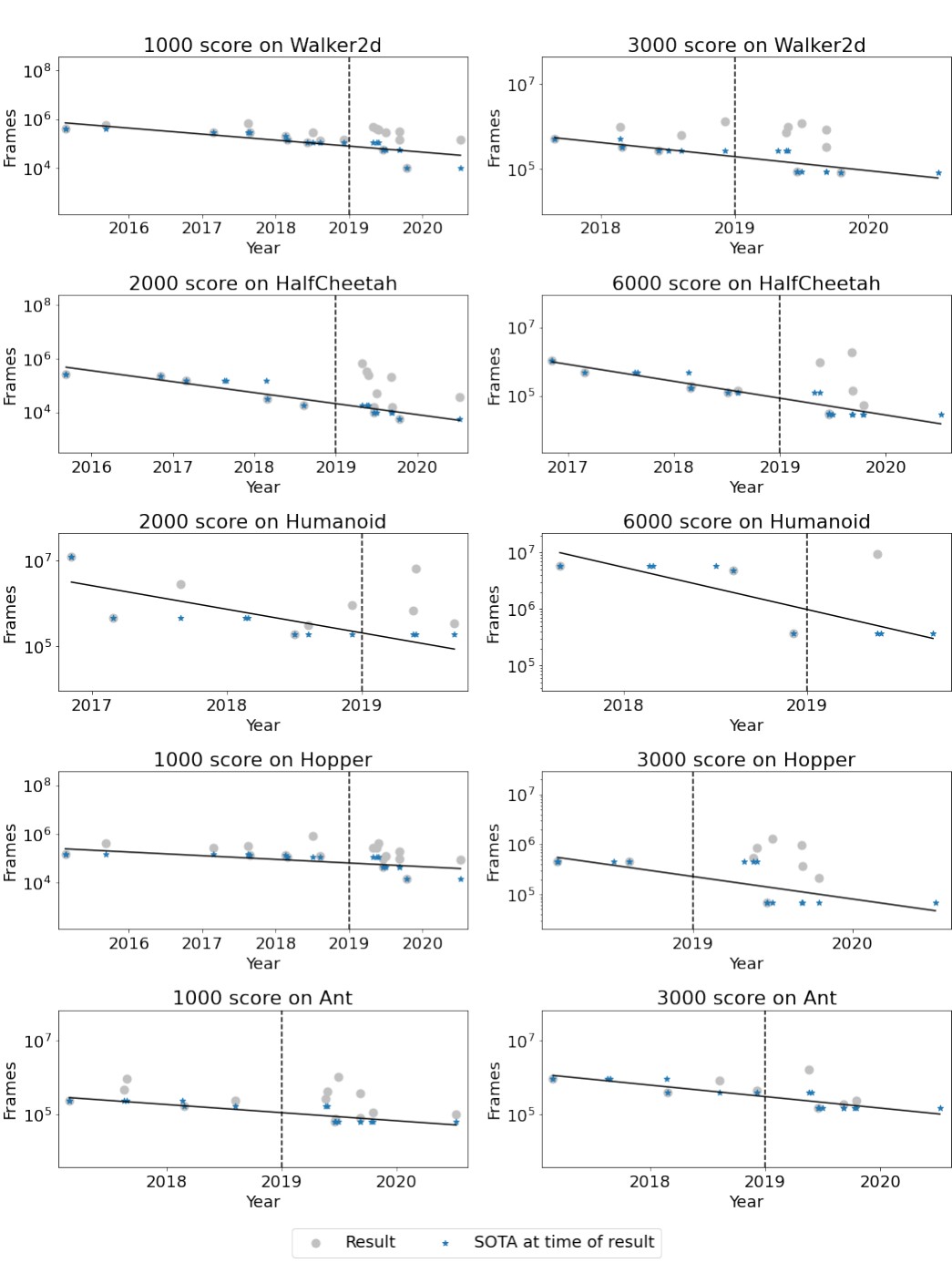

Figure 5: Frames needed to reach the specified score for various MuJoCo tasks from OpenAI gym over time. The vertical line indicates the switch from gym-v1 results to gym-v2 results and results from ambiguous versions of gym. Graphs are normalized such that graphs with the same slope indicate the same doubling time, despite different measurement horizons. The black line is the best fitting exponential model for the SOTA (all SOTA).

| Task and Score | Doubling time (all SOTA) | Doubling time (jumps only) | Doubling time (continuous until 10.08.2020) |
|---|---|---|---|
| Walker2D 1000 | 14.6 months | 12.1 months | 11.8 months |
| Walker2D 3000 | 10.9 months | 9.1 months | 10.8 months |
| HalfCheetah 2000 | 8.8 months (*) | 8.1 months (*) | 8.3 months (*) |
| HalfCheetah 6000 | 7.3 months | 6.5 months | 7.8 months |
| Humanoid 2000 | 6.5 months | 4.2 months | 11.1 months |
| Humanoid 6000 | 4.8 months | 4.8 months | 6.1 months |
| Hopper 1000 | 24 months | 19.3 months | 20.3 months |
| Hopper 3000 | 8 months | 5.7 months | 7.2 months |
| Ant 1000 | 16.4 months (*) | 14.5 months (*) | 16.9 months (*) |
| Ant 3000 | 11.6 months | 10.4 months | 12.1 months |

Table 4: Estimated doubling times in sample efficiency for various MuJoCo tasks from OpenAI gym. (*): We did exclude some results that might have improved the SOTA due to insufficient resolution around zero frames or a y-axis cutoff.

| Task and Score | Doubling time (all SOTA) | Doubling time (jumps only) | Doubling time (continuous until 10.08.2020) |
|---|---|---|---|
| Walker Walk 400 | 4.5 months (*) | 4.2 months (*) | 4 months (*) |
| Walker Walk 600 | 3.9 months | 4.4 months | 3.3 months |
| Walker Walk 800 | 3.8 months | 4.3 months | 3.2 months |
| Cheetah Run 400 | 3.7 months | 3.6 months | 3.2 months |
| Cheetah Run 600 | 8.5 months | 7.5 months | 8 months |
| Cheetah Run 800 | 9.2 months | 6.4 months | 7.8 months |
| Ball in Cup Catch 400 | 4 months | 4.1 months | 3.5 months |
| Ball in Cup Catch 600 | 3.9 months | 3.8 months | 3.5 months |
| Ball in Cup Catch 800 | 3.8 months | 4 months | 3.3 months |
| Finger Spin 400 | 4 months (*) | 4 months (*) | 3.6 months (*) |
| Finger Spin 600 | 4.1 months (*) | 3.6 months (*) | 3.6 months (*) |
| Finger Spin 800 | 4.3 months | 4.4 months | 4.2 months |

Table 5: Estimated doubling times in sample efficiency for various DeepMind Control Suite tasks (on pixels). (*): We did exclude some results that might have improved the SOTA due to insufficient resolution around zero frames or a y-axis cutoff.

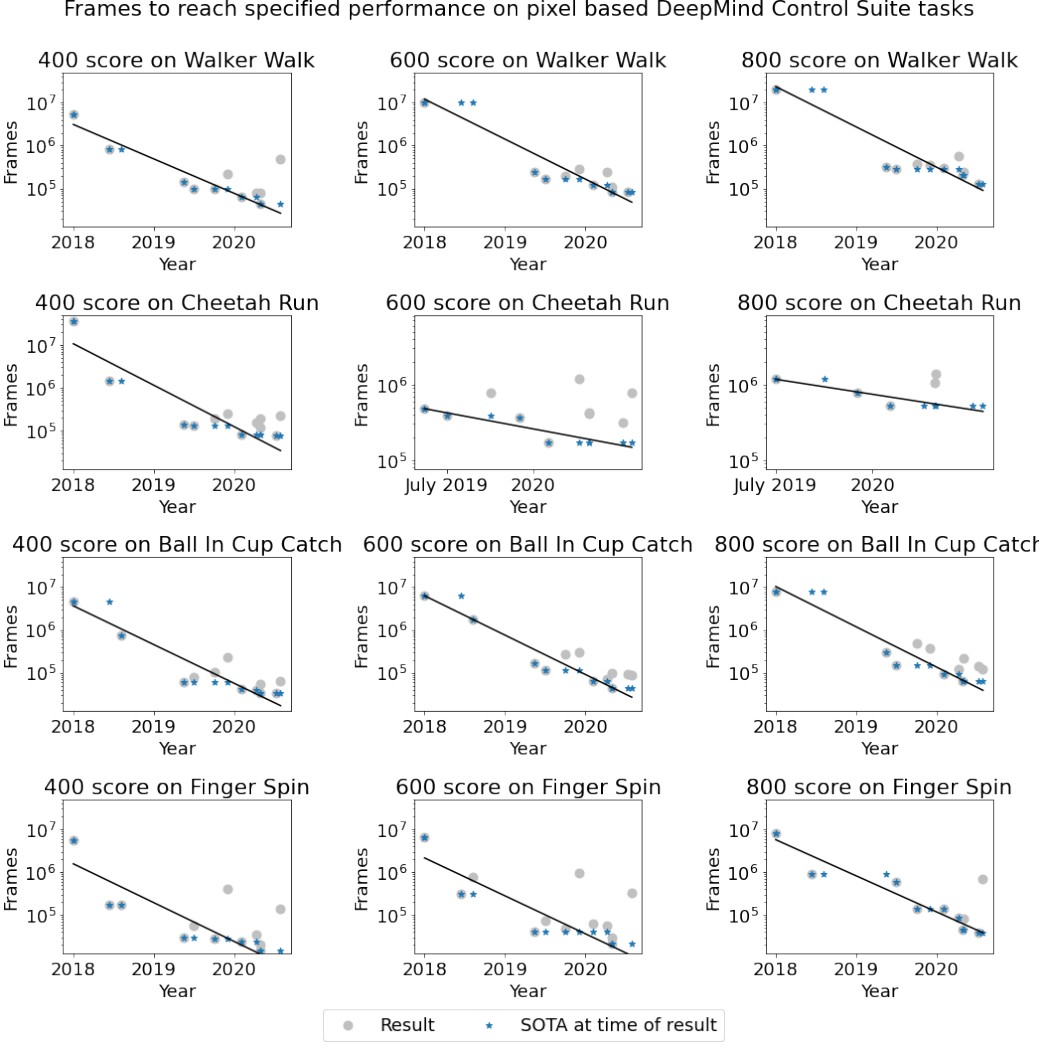

Figure 6: Frames needed to reach the specified score on various DeepMind Control Suite tasks over time. Graphs are normalized such that graphs with the same slope indicate the same doubling time, despite different measurement horizons. The black line is the best fitting exponential model for the SOTA (all SOTA).

