# OpenReview forum: "Measuring Progress in Deep Reinforcement Learning Sample Efficiency "
_ICLR.cc/2021/Conference — Reject_

### Official Review · AnonReviewer2 · 2020-10-22
**A meta-analysis of sample efficiency in deep RL**

**Rating:** 4
**Confidence:** 4

**Review:**

##################################

Summary:
This paper conducts a meta-analysis of the trend in sample efficiency in deep RL. The authors argue that this is an informative measure of the progress in the field, in addition to the usual metrics of reward for given tasks, as it is an important consideration when applying deep RL to real world problems. They measure sample efficiency by the number of samples from the environment that is needed to achieve some reward. The amount of time to double sample efficiency is computed for Atari and continuous control environments with state or pixel input.

##################################

Pros:
1. In deep RL, usually evaluation is done by reporting the reward achieved for some set of environments S after some number of steps N during training. Since S and N vary between papers, it is difficult to measure progress in this field. By using sample efficiency as a metric, this paper is able to give a retrospective of the progress for environments with pixel input, which appears to be missing from the literature.
2. The paper is able to estimate the fraction of progress due to 1) increased sample sizes 2) algorithmic improvements. I found this to be the most interesting conclusion from the paper.
3. Based on the history of the algorithms, the paper provides discussion of the main breakthroughs, which is very helpful.

Cons:
1. On the technical side, the paper does not seem to have much novelty. The analysis approach is the same as that of Hernandez & Brown (2020), with minor changes to adapt to the deep RL setting.
2. There are quite a few limitations to the analysis, which the authors discuss in Section 5 and Appendix D. My main concern is about the hyper-parameters. Deep RL is notoriously sensitive, and the way that researchers select hyper-parameters can have great effect on the final results. Therefore, I am concerned that differences in the amount of effort used to tune hyper-parameters may either mask improvements in sample efficiency or lead to spurious conclusions that sample efficiency has improved. Could the authors discuss how this issue could affect the conclusions in the paper?

##################################

Overall:
I am not sure that this work in its current form is the right fit for ICLR. It does not have much technical novelty, and its reliance on reported training curves, in my opinion, decreases the credibility of the results. In addition, since it is a meta-analysis, I would suggest that a venue like the "ML Retrospectives, Surveys & meta-Analyses (ML- RSA)" workshop at NeurIPS would be a better fit.

##################################

Further comments and questions:
1. I think a more in-depth discussion of the factors leading to progress in sample efficiency, currently in Appendix C, would be very helpful to researchers thinking about new ideas. Therefore, I suggest that it be moved to the main paper.
2. In Section 4.2, how large is the variation in the amount of frames used in the unrestricted benchmark, and do most of those algorithms use >>200M frames? I think that a lot of variation in the amount of frames would imply a less accurate fit of the trend.

##################################

Update after reading other reviews and author responses:

Thanks to the authors for answering my questions. However, my main concerns were not addressed, so I will keep my score.

---

> ### Author Response · Authors · 2020-11-15
> **Reply**
>
> Thank you for your time!
>
> We moved Appendix C into the main text. This also includes a paragraph on hyperparameters. We also added another comment on how hyperparameter tuning might correspond to unsustainable short term progress near the end of the discussion.
>
> The first five results in the unrestricted benchmark use 200M Frames, the next 250M and the last three use 22.8B, 55.3B and 20B frames. As we mention in the paper, the results are in 4.2. are especially uncertain because of the low amount of data points compared to the other analyses.

---

### Official Review · AnonReviewer1 · 2020-10-28

**Rating:** 5
**Confidence:** 5

**Review:**

The paper is trying to make extensive and systematic investigation in deep RL papers to measure the recent progress in the broad literatures. The authors look into published state-of-the-art results in Atari, state-based continuous control and pixel-based continuous control. Inferring from papers' published training curves, the authors estimated and found sample efficiency doubling time is 10-18 months for Atari, 5-24 months for continuous control and 4-9 months for pixel-based control tasks.

Pros:
+ The reviewer really likes the extensive efforts put on the paper, which includes deep survey on published DRL literatures, analysis on progress of agents, open discussions on limitations of the analysis in Appendix part D, etc;
+ The authors state super clearly about their methodology in the study in Section 3, where they explain how they measure the sample efficiency, how they select publications and SOTA result for analysis and what's the source of numbers used for analysis;

Cons:
- The biggest concern the reviewer has, which is also discussed and stated in Appendix D, is that many of the published papers, especially those on Atari, were tuned and focused on the performance *score* at 200M frames (or other frame limits), instead of optimising for data efficiency. In fact, as also be mentioned in the paper, van Hasselt et al. 2019 and Kielak 2020 noted that if you tune  the hyperparms for data efficiency, you may draw a very different conclusion. At the same time, the reviewer acknowledged that reproducing all RL results and tuning them separately for data efficiency would require dramatic amount of time and compute resources;
- Related to the above point, when taking the training curve as reference to plot the data efficiency progress figures like Figure 1, It may be ok for the 200M frames evaluation agents, but when it comes to high frame regimes which R2D2 or MuZero uses, they were aiming for absolute performance score instead of aiming for data efficiency. IMO, it's hard to draw conclusions on data efficiency with those agents which consume billions of frames and they didn't consider data efficiency;
-  The continuous control tasks might be a better place to conduct study on data efficiency where more data efficiency related publications go into and those literatures and applications care much more about data efficiency than the ones on Atari. The reviewer found that the survey put more effort and content on the Atari results instead of the control tasks could be improved;
- The reviewer would love to see more discussions like the Appendix part C on how different agents improved data efficiency. It might be worth expanding these sections and put those into the main paper instead of the Appendix. Answering how DRL has progressed on data efficiency is an essential addition to measuring how much it has progressed;



Minor:
- For Figure 1 (a) some agent was labelled at a wrong published year, e.g. FRODO was published in 2020 instead of 2018.

---

> ### Author Response · Authors · 2020-11-15
> **Reply**
>
> Thank you for your time!
>
> We agree that agents like MuZero and R2D2 cannot be used for conclusions about data efficiency and did not aim to do so. If you told us, where you got the impression that we did, we are happy to clarify this in the paper.
>
> We have moved appendix C to the main paper and will try to further improve on the section. We will also consider, how we can expand on our analysis for continuous control.
>
> We corrected the date for FRODO and double checked all other dates in Figure 1 (a)

---

> > ### Comment · AnonReviewer1 · 2020-11-24
> > **Reply**
> >
> > Thanks for the updates! Unfortunately I'll keep the original score.

---

### Official Review · AnonReviewer3 · 2020-11-02
**Not enough material for a paper**

**Rating:** 2
**Confidence:** 5

**Review:**

This paper aims to measure sample efficiency of available RL  methods instead of algorithmic improvement.  It draws a conclusion about continuing improved RL sample efficiency in the past few years.  This conclusion is rather thin to fill a complete paper.
And the evaluation procedure is rather limited: it directly uses data reported from the original papers without re-producing them at the same environment. These different papers came across a span of at least four years;  although Mujoco is a standard benchmark,  it also has evolved multiple times, basically, there are so many uncontrolled variables in the plot reported by authors. I consider these results not eligible for interpretation. And the algorithms considered here for sample efficiency are not on the same ground, some are off-policy, some are on-policy, off-policy algorithms are naturally far more sample efficient than on-policy, they can't be compared in this way.  Also, authors only report results for score 400 and 2000, which is also inadequate.    In the pixel experiments, it mentions SLAC and dreaming, they are not purely pixel-based RL, rather a latent space is trained at first; so it's so a valid comparison between them and methods like CuRL. In the state experiments, SLAC is not purely state-based, it trains VAE first for a low-dim latent space.

In all, the material presented in this paper does not fill a complete paper, and the evaluation protocol and conclusions are not valid.

---

> ### Author Response · Authors · 2020-11-15
> **Reply**
>
> Thank you for your time!
>
> We only used MuJoCo results from gym-v1 up to 2019 and see no indication that there is a discontinuity at the point in time, where multiple versions are taken into consideration for the first time.
>
> We agree that progress in sample efficiency was largely driven by off-policy algorithms. However, we are of the opinion that the availability of competitive off-policy algorithms for continuous control represents progress in sample efficiency compared to a time, where such algorithms were not available, such that comparing on-policy algorithms to off-policy algorithms (and later model-based algorithms) does not present a problem. Similarly, we are of the opinion that for a broad evaluation of progress in DRL sample efficiency, algorithms training a latent space first can be compared to algorithms that don’t do this, as long as the frames used for training the latent space are counted towards the amount of training samples. We are under the impression that this was done for Dreaming and SLAC and added more detailed information to appendix B.
>
> Lastly, we did in fact report results for three different scores on four different DeepMind control suite tasks and two different scores for five different MuJoCo/gym tasks in the appendix. We added a reference to the appendix in a footnote in the relevant section to make this more clear.

---

### Official Review · AnonReviewer4 · 2020-11-03

**Rating:** 5
**Confidence:** 5

**Review:**

Summary:

The paper proposes to retrospectively benchmark the sample-efficiency on widely used simulated deep RL benchmarks such as Atari and DMControl across the years. The paper shows some interesting trends with respect to how both the algorithmic improvements as well as the use of increased number of frames have driven the progress in the SoTA scores reported on the Atari HNS benchmark. The paper also shows that there has been exponential progress in the sample-efficiency on both Atari and DMControl with nice log-linear plots. This is an interesting analysis, on the lines of OpenAI's papers on Scaling Laws, but done for Deep RL.

Pros:

Useful review article for the community.
Communicates progress in very visible and scientifically precise manner.
Disentangles progress made in algorithmic efficiency and using more frames (distributed training), and wall clock time reduction.
Cons:

The DMControl results could be presented better.. Specific points below:
DeepMind Control results: (i) In DMControl plots, using SLACv2 and other methods on the same plot is incorrect because SLACv2 performs more gradient updates per batch while other methods do not and it has been known that it is a contributing factor to sample efficiency. The author is encouraged to check an ablation in CURL Appendix E.2 where CURL is shown to be superior to SLAC-v2 on 3/4 envs when using similar update frequencies for 500k, and 2/4 for 100k steps respectively. (ii) Performance numbers used for CURL have been taken from DrQ which do not reflect the accurate numbers. Based on the numbers reported in RAD, for both CURL and RAD, the performance for Walker from CURL seems to be at 403 +/- 24, while DrQ reports 344 +/- 132. Would recommend taking numbers directly from updated versions of CURL/RAD on arXiv which are consistent with each other. Would also recommend adding multiple environments to do the analysis on DMControl as opposed to just one arbitrary hand-picked environment without a clear justification. (iii) As pointed out in DrQ, SLAC uses 100k exploration steps which are not counted in the reported sample-efficiency values. Check Table 1 in DrQ. I believe this is an issue with SLACv2 as well.

The paper should talk a little more detail about how the hardware has changed the game in terms of wall clock time reduction, what kind of hardware, etc. A plot wrt computation flops would be useful to have.

The amount of actionable insight gathered from the paper - in terms of - after reading this paper, what can I do to make my DRL systems more efficient? - is not clear to me. I would be curious if the authors can explain this to me given there's a chance I missed it.

Rating: Weak Reject - I believe the paper has a lot of room for improvement and has to provide some strong good insights out of the analysis, but I am open to updating my views based on author response.

---

> ### Author Response · Authors · 2020-11-15
> **Reply**
>
> Thank you for your time!
>
> We moved appendix C, which summarizes important contributors to improved sample efficiency to the main paper. We will try to further improve on the section. We also added a sentence about the role of hardware in the clock time speedups.
>
> Regarding (i), We agree that the number of used gradient steps could be an issue, in fact is a special case of the general issue with hyperparameters that is discussed in the appendix and section 5.4 (which was moved from the appendix). However, even if SLAC was using multiple gradient steps, we believe that excluding algorithms for tuning their hyperparameters for sample efficiency would distort results even more, as algorithms designed for strong sample efficiency are also most likely to tune their hyperparameters for sample efficiency, especially in a domain where an increasing amount of papers specifically target sample efficiency.
>
> Also, all versions of SLAC seem to report using a single gradient step per environment step for the DM Control Suite. Arxiv versions 1 to 3 also report a single step for gym, while version 4 reports using three steps on gym. As the reported results change for the Control Suite but not gym between v3 and v4, this might be a typo. We neither considered version 3 or 4, as both were published after our study’s cutoff date. If you are aware of whether the CURL paper refers to arxiv version 2 or a later version with “SLACv2“, we would be grateful for that information. As far as we are aware, there are no training curves/fine grained results available for the version of CURL that uses 3 gradient steps, but would be more than happy to include the results in our analysis if that data was to be published. However, note that the results with 3 and 1 gradient steps seem quite close for both 100k and 500k frames, in most cases, which is also evidence that the effect of additional gradient steps might be quite limited.
>
> Regarding (ii), we changed the results for CURL to the results from Figure 8 in the original publication. Thank your for pointing out the issue with the numbers from DRQ. We in fact report results for three different scores on four different DeepMind control suite tasks and two different scores for five different MuJoCo/gym tasks in the appendix and also refer to them in our analysis. We added a reference to the appendix in a footnote in the relevant section to make this more clear.
>
> Regarding (iii): This is tricky: Arxiv versions 1 and 2 of SLAC don’t report using these exploratory steps and the pseudocode algorithm also does not include them. Versions 3 and 4 report 10k agent steps but claims they are included in the plots. We added a more detailed description of this problem to the appendix and also performed a sensitivity analysis that showed that the effect of adding 10000 environment steps to all SLAC results changes doubling times by at most 0.3 months (and a lot less for most environment/task combinations)

---

### Decision · Program_Chairs · 2021-01-07
**Final Decision**

**Decision:**

Reject

**Comment:**

Although all reviewers agree that this is an interesting analysis of sample efficiency in Deep RL over the past few years, there is also a consensus that it is not enough material for an ICLR paper. I also share this sentiment, which motivates the "Reject" decision. This work could have been made stronger by reproducing previous results (even partially) and sharing the corresponding code, so as to provide a fair and controlled comparison of algorithms, and setting the stage for future progress in this area. In its form, it is better suited for a presentation at a workshop than for the main conference.